

**Thermodynamics of Saline and Fresh Water Mixing in Estuaries**
Zhilin Zhang and Hubert H.G. Savenije
Delft University of Technology
**Abstract**
Mixing of saline and fresh water is a process of energy dissipation. The fresh water flow that enters
an estuary from the river contains potential energy with respect to the saline ocean water. This
potential energy is able to perform work. Looking from the ocean to the river, there is a gradual
transition from saline to fresh water and an associated rise of the water level in accordance with the
increase of potential energy. Alluvial estuaries are systems that are free to adjust dissipation
processes to the energy sources that drive them, primarily the kinetic energy of the tide and the
potential energy of the river flow, and to a minor extent the energy in wind and waves. Mixing is
the process that dissipates the potential energy of the fresh water. The Maximum Power (MP)
concept assumes that this dissipation takes place at maximum power, whereby the different mixing
mechanisms of the estuary jointly perform the work. In this paper, the power is maximized with
respect to the dispersion coefficient that reflects the combined mixing processes. The resulting
equation is an additional differential equation that can be solved in combination with the advection-
dispersion equation, requiring only two boundary conditions for the salinity and the dispersion. The
new equation has been confronted with 52 salinity distributions observed in 23 estuaries in different
parts of the world and performed very well, even better than the well-tested empirical Van der
Burgh equation that required a calibration parameter, which with this equation is no longer needed.

**1. Introduction**
Mixing of fresh and saline water in estuaries is governed by the dispersion-advection equation,
which results from the combination of the salt balance and the water balance under partial to well-
mixed conditions (see e.g. Savenije, 2005). The partially to well-mixed condition applies when the
increase of the salinity over the depth is gradual. The salinity equation reads:

$$A_s \frac{\partial S}{\partial t} + Q\frac{\partial S}{\partial x} - \frac{\partial}{\partial x}\left( AD\frac{\partial S}{\partial x} \right) = 0 \qquad (1)$$
Here, $S$ [psu] is the salinity of the water, $Q$ [L$^3$T$^{-1}$] is the water flow in the estuary, $A$ [L$^2$] is the
cross-sectional area of the flow (not necessarily equal to the storage cross-section $A_S$), and $D$ [L$^2$T$^{-1}$]
is the dispersion coefficient. The first term reflects the change of the salinity over time as a result of
the balance between the advection by the water flow (second term) and the mixing of water with
different salinity by dispersive exchange flows (third term). If there is no other source of salinity,
then the sum of these terms is zero. If we average this equation over a tidal period, then the first
term reflects the long term change of the salinity as a result of the balance between the advection of
fresh water from the river, and the tidal average exchange flows. In a steady state, where the first
term is zero, the equation can be simply integrated with respect to $x$, yielding:

$$Q\left(S - S_f\right) - AD\frac{\partial S}{\partial x} = 0 \qquad (2)$$
with the condition that at the upstream boundary $\partial S / \partial x = 0$ and $S=S_f$, the salinity of the fresh river
water. In the steady state situation the discharge $Q$ then equals the fresh water discharge coming
from upstream, which has a negative value moving seaward; similarly the salinity gradient is
negative with the salinity decreasing in upstream direction. Assuming that in a given estuary the



geometry $A(x)$ is known, as well as the observed salinity and discharge of the fresh river water, then
this differential equation has two unknowns $D(x)$ and $S(x)$.
In the steady state salt balance equation the flushing out of salt by the fresh river discharge is
balanced by the exchange of saline and fresh water resulting from a combination of mixing
processes that cause an upriver flux of salt. The sketch in Figure 1 presents the system description
with a typical longitudinal salinity distribution (in red). It also shows the associated water level (in
blue), which has an upstream gradient due to the decreasing salinity. Because of the density
difference, the hydrostatic pressures on both sides (in yellow) are not equal. The water level at the
toe of the salt intrusion curve is $\Delta h$ higher, resulting in a seaward pressure difference near the
surface and an inland pressure difference near the bottom. Although the hydrostatic forces (the
integrals of the hydrostatic pressure distributions) are equal and opposed in steady state, they have
different working lines, a distance $\Delta h/3$ apart. This triggers an angular momentum, which drives the
gravitational circulation.

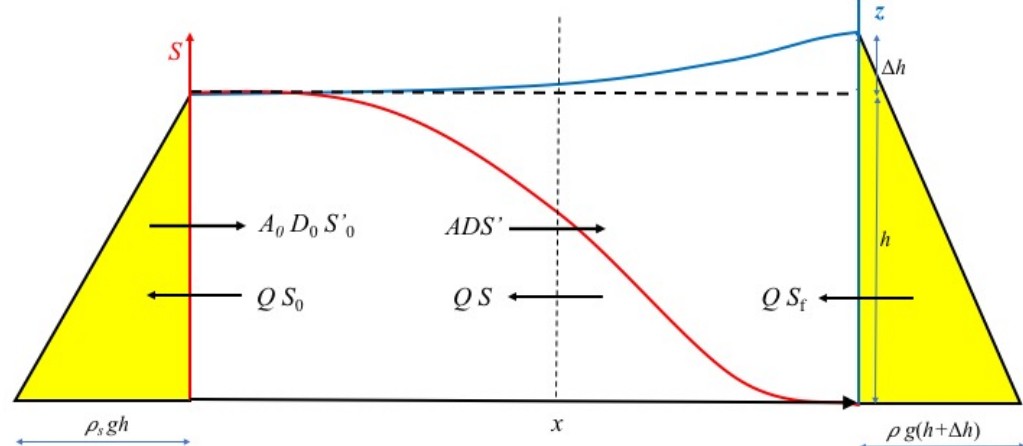

**Figure 1. System description of the salt and fresh water mixing in an estuary, with the seaside on the left and the river side on**
**the right. The water level (blue line) has a slope as a result of the salinity distribution (red line). In yellow are the hydrostatic**
**pressure distributions on both sides. The black arrows show the boundary fluxes.**
The dispersion coefficient is generally determined by calibration on observations, or predicted by
(semi-)empirical methods. Providing a theoretical basis for the dispersion coefficient is not trivial.
A fundamental question is what this dispersion actually is. Is it a physical parameter, or merely a
parameter that follows from averaging the complex turbulent flow patterns in a natural watercourse.
MacCready (2004), for instance, was able to derive an analytical expression for the dispersion as a
function of the salinity gradient and geometric, hydraulic and turbulence parameters. But also this
derivation required simplifying assumptions.
The complication is that there are many different mixing processes at work. One can distinguish:
tidal shear, tidal pumping, tidal trapping, gravitation circulation (e.g. Fischer et al., 1979) and
residual circulation due to the interaction between ebb and flood channels (Nguyen and Savenije,


2008; Zhang and Savenije, 2017). And these different processes can be split up in many
subcomponents. Park and James (1990), for instance, distinguished 66 components, grouped into 11
terms. This reductionist approach, unfortunately, did not lead to more insight.
**2. Applying thermodynamics to salt and fresh water mixing**
Here we take a system's approach, where the assumption is that the different mechanisms are not
independent but are jointly at work to reduce the salinity gradient that drives the exchange flows.
We use the concept of Maximum Power, as described by Kleidon (2016). Kleidon defines Earth
system processes as dissipative systems that do conserve mass and energy, but export entropy.
These systems tend to function at maximum power, whereby the power of the system can be
defined as the product of a process flux and the gradient driving the flux. The ability to maintain
this power (i.e., work through time) in steady state results from the exchange fluxes at the system
boundary, and when work is performed at the maximum possible rate within the system
("Maximum Power"), this state reflects the conditions at the system boundary. The key parameter
describing the process can then be found by maximizing the power.
From an energy perspective, we see that the fresh water flux, which has a lower density than saline
water and, without a counteracting process, would float on top of the saline water, adds potential
energy to the system, while the tide, which flows in and out of the estuary at a regular pace, creates
turbulence, mixes the fresh and saline water and hence works at reducing this potential energy. This
is why dispersion predictors are generally linked to the Estuarine Richardson number, which
represents the ratio of the potential energy of the fresh water entering the estuary to the kinetic
energy of the tidal flow.
In thermodynamic terms, the fresh water flux maintains a potential energy gradient, which triggers
mixing processes that work at depleting this gradient. Because the strength of the mixing of fresh
and saline water in turn depends on this gradient, there is an optimum where the mixing process
performs at maximum power. From a system point of view, it is not really relevant which particular
mixing process is dominant, or how these different processes jointly reduce the salinity gradient.
What is relevant is how the optimum flux associated with this mixing process, yielding maximum
power, depends on the dispersion.
In our case, the power derived from the potential energy of the fresh water flux is described by the
product of the upstream dispersive water flux and the gradient in geopotential height driving this
flux, or alternatively, the product of the dispersive exchange flux and the water level gradient.
The water level gradient follows from the balance between the hydrostatic pressures of fresh and
saline water (see e.g. Savenije, 2005), resulting in:
$$\frac{\partial z}{\partial x} = -\frac{h}{2\rho}\frac{\partial \rho}{\partial x} \qquad (3)$$
where $z$ [L] is the tidal average water level, $h$ [L] is the tidal average water depth and $\rho$ [ML$^{-3}$]is the
density of the saline water. Note that this equation applies to the case where the river flow velocity
is small, which is the case when estuaries are well mixed. Otherwise a backwater effect should be
included, but this only applies to a situation of high river discharge when the salt intrudes by means
of a salt wedge with a sharp interface.
One can express the density of saline water as a function of the salinity $S$ [psu]: $\rho = 1000 + \alpha_1 S$
(kg/m$^3$) where $\alpha_1$ is a constant with a value of about 25/35, because seawater with a salinity of 35
psu has a density of about 1025 kg/m$^3$. As a result, eq.(3) can be written as:





$$\frac{\partial z}{\partial x} = -\alpha_1 \frac{h}{2\rho} S'$$ (4)
The upstream dispersive flux is implicit in the salt balance equation, which in steady state can be
written as:

$$Q(S - S_f) = ADS'$$ (5)
where $S'$ [L$^{-1}$] is the salinity gradient, which is negative in upstream direction. So the left hand term
is the salt flux due to the fresh water of the river that pushes back the salt, whereas the right hand
term is the dispersive intrusion of salt due to the exchange flux of the combined mixing processes
(see Figure 1). Writing both sides as water fluxes results in:
$$Q = \frac{ADS'}{(S - S_f)}$$ (6)
The right hand side is the water exchange flux, which is the flux that depletes the gradient. As (6)
shows, in steady state this exchange flux is equal to the fresh water discharge. Combination of the
flux and the gradient leads to the power of the mixing system per unit length (defined as a positive
quantity):
$$P = -\rho g Q \frac{\partial z}{\partial x} = \alpha_1 Q \frac{gh}{2} S'$$ (7)
To apply the theory of maximum power to the dispersive process, we need to maximize the power
with regard to the dispersion coefficient, which is the parameter representing the mixing and which
is the main unknown in salt intrusion prediction:
$$\frac{\partial P}{\partial D} = 0$$ (8)
Applying (8) with constant river discharge $Q$ and constant depth $h$ -- the property of an ideal
alluvial estuary, according to Savenije (2005) -- leads to:
$$\frac{\partial S'}{\partial D} = 0$$ (9)
Using the salt balance equation, where $S' = Q(S - S_f)/(AD)$, differentiation leads to:
$$\frac{S'}{(S - S_f)}\left\{ \frac{S'}{D'} - \frac{A'(S - S_f)}{AD'} - \frac{(S - S_f)}{D} \right\} = 0$$ (10)
The solution $S'=0$ is trivial. For non-zero salinity gradient, the solution is:
$$\frac{DS'}{(S - S_f)D'} = \frac{A'D}{AD'} + 1$$ (11)
We introduce three length scales: $a = -(A - A_f)/A'$, $s = -(S - S_f)/S'$ and $d = -D/D'$, where $a$ is
the convergence length of an exponentially varying estuary cross-section which tends towards the
cross-section of the river $A_f$, $s$ is length scale of the longitudinal salinity variation, and $d$ is length
scale of the longitudinal variation of dispersion. In macro-tidal estuaries, the part of the estuary
where the salt intrusion occurs has a much larger cross-section than the upstream river, such that
$A_f \ll A$ and $a \approx A/A'$. In riverine estuaries, where this is not the case, a factor $\varepsilon = (1 - A_f/A)$ should be
included. All length scales have the dimension of [L]. In an exponentially shaped estuary, the
convergence length is a constant, but $d$ and $s$ vary with $x$. It can be shown that the proportion $s/d$
equals the Van der Burgh coefficient $K = AD'/Q$, which in this approach varies as a function of $x$,





although generally assumed constant (e.g. Savenije, 2005; and Zhang and Savenije, 2017). Using
these length scales, eq. (11) can be written as:
$\dfrac{s}{d} = \dfrac{a}{a + d\varepsilon}$     (12)
or:
$s = \dfrac{ad}{a + d\varepsilon}$     (12a)
or:

$d = \dfrac{as}{a - s\varepsilon}$    (12b)

where in estuaries with a pronounced funnel shape $\varepsilon \approx 1$. Eq.(12) is an additional equation to the salt
balance, which in terms of the length scales reads: $s = -AD/Q$. As a result, we have two
differential equations with two unknowns: $S(x)$ and $D(x)$. Adding two boundary conditions at a
given point $x=0$: $S_0$ and $D_0$ would solve the system. The first boundary condition is simply sea
salinity if the boundary is chosen at the estuary mouth. Then the only unknown parameter left is the
value for the dispersion at the ocean boundary. For this boundary value empirical predictive
equations have been developed which relate the $D_0$ to the Estuarine Richardson number (e.g. by
Gisen et al., 2015), which goes beyond this paper. If observations of salinity distributions are
available, then the value of $D_0$ is obtained by calibration.

What the maximum power equation has contributed is that it provides an additional equation. In the
past, a solution could only be found if an empirical equation was added describing $D(x)$, containing
an additional calibration parameter. In the approach by Savenije (2005) this was the empirical Van
der Burgh equation containing the constant Van der Burgh coefficient $K$. However, with the new
equation (12), which in fact represents a spatially varying Van der Burgh coefficient, this additional
calibration parameter is no longer required. So this new approach replaces and empirical equation
for a physically based equation and reduces the number of calibration parameters to one: the
dispersion at a well-chosen boundary condition.

**3. Application**
The two equations (2) and (12) together can be solved numerically by a simple linear integration
scheme. As boundary condition it requires values for $S(x_1)$ and $D(x_1)$ at a well-chosen location $x=x_1$.
In alluvial estuaries the cross-sectional area $A(x)$ generally varies according to an exponential
function which often has an inflection point (see for example Figure 2 describing the Maputo
Estuary in Mozambique). The boundary condition is best taken at this inflection point if the estuary
has one. If the estuary has no inflection point, as is the case in the Limpopo estuary (see Figure 3),
then the boundary condition is taken at the estuary mouth.

The downstream part of estuaries with an inflection point has a much shorter convergence length,
giving the estuary a typical trumped shape. This wider part is generally not longer than about 10
km, which is the distance over which ocean waves dissipate their energy. Beyond the inflection
point, the shape is determined by the combination of kinetic energy of the tide and the potential
energy of the river flow. If the tidal energy is dominant over the potential energy of the river, then
the convergence is short, leading to a pronounced funnel shape; if the potential energy of the river is
large due to regular and substantial flood flows, then the convergence is large, typical for deltas.
Hence, the topography can be described by two branches:

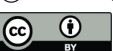

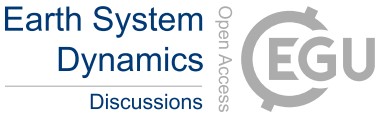

$$A = A_f + (A_0 - A_f)\exp(-x/a_0)\,\text{if}\,0 < x < x_1$$
$$A = A_f + (A_1 - A_f)\exp(-(x-x_1)/a_1)\,\text{if}\,x \geq x_1$$
(13)

where $A_0$ and $A_1$ are the cross-sectional areas at $x=0$ and $x=x_1$, respectively, and $a_0$ and $a_1$ are the
convergence lengths of the lower and upper segments. In some cases, where ocean waves don't
penetrate the estuary, there is no inflection point and $x_1=0$. The Maputo (see Figure 2) has two
segments, whereas the Limpopo Estuary (see Figure 3), an estuary in Mozambique 200 km north of
the Maputo semi-closed by a sand bar, has a single branch. It can also be seen that in the Limpopo
the size of the river cross-section is not negligible and that $\varepsilon<1$ showing a slight curve in the
exponential functions.
Subsequently we have integrated the equations (2) and (12) conjunctively by a simple explicit
numerical scheme in a spreadsheet and confronted the solution with observations. The solutions are
fitted to the data by selecting values for $S$ and $D$ at the boundary condition $x=x_1$ (or at $x=0$ for the
Limpopo). Figures 4 and 5 show applications of the solution to selected observations in the Maputo
and Limpopo estuaries. In the supplementary material more applications are shown, also for other
estuaries in different parts of the world.






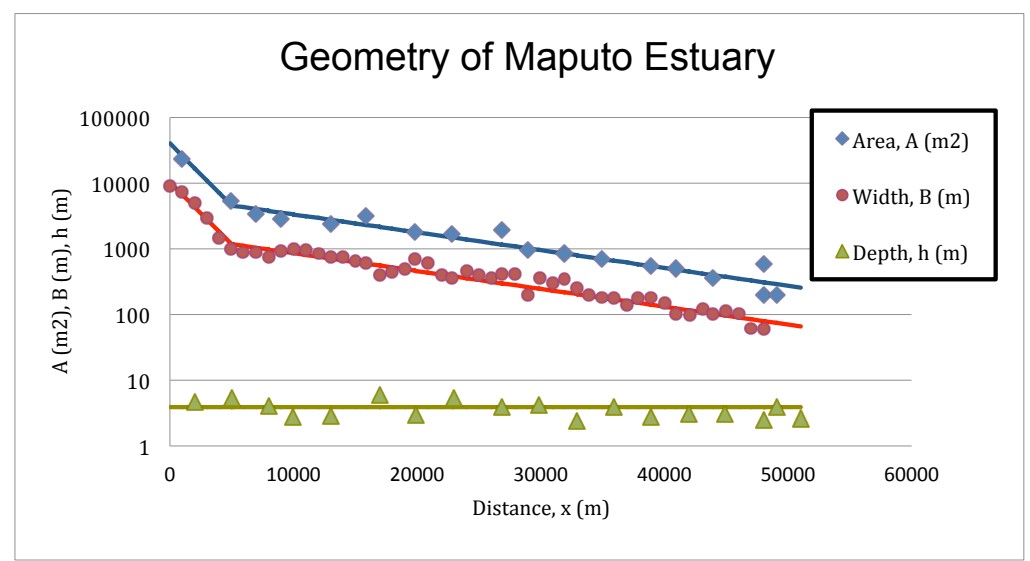

**Figure 2, Geometry of the Maputo Estuary, showing the cross-sectional area *A* (blue diamonds), the width *B* (red dots) and**
**the depth *h* (green triangles) on a logarithmic scale, as a function of the distance from the mouth. The inflection point at**
**x1=5000 m separates the lower segment with a convergence length of $a_0$=2300 m from the upper segment with $a_1$=16000m.**

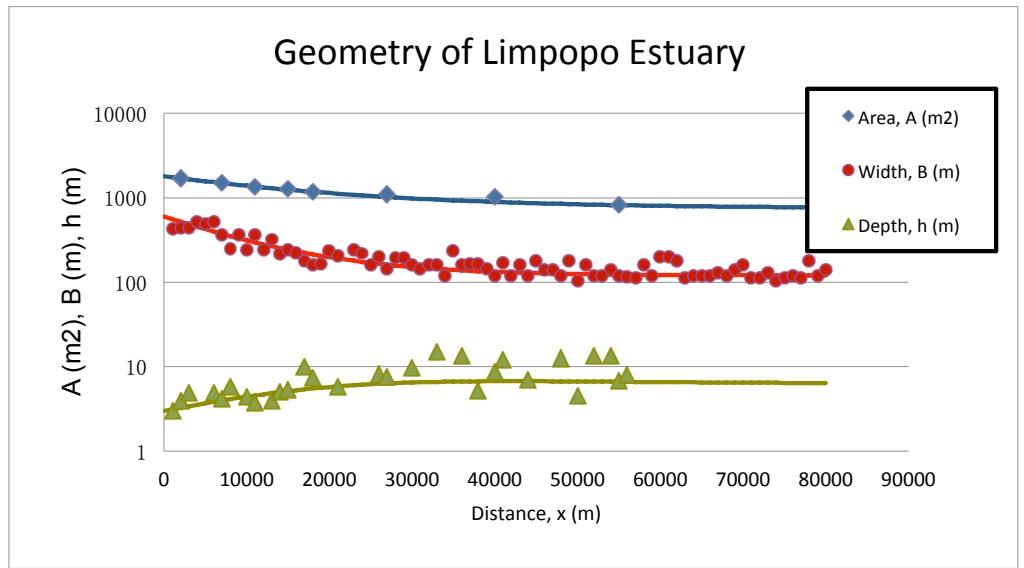

**Figure 3. Geometry of the Limpopo Estuary, showing the cross-sectional area *A* (blue diamonds), the width *B* (red dots) and**
**the depth *h* (green triangles) on a logarithmic scale, as a function of the distance from the mouth. There is no inflection point,**
**but the estuary converges exponentially towards the river cross-section Af= 800 m$^2$, with a convergence length of 20 km.**




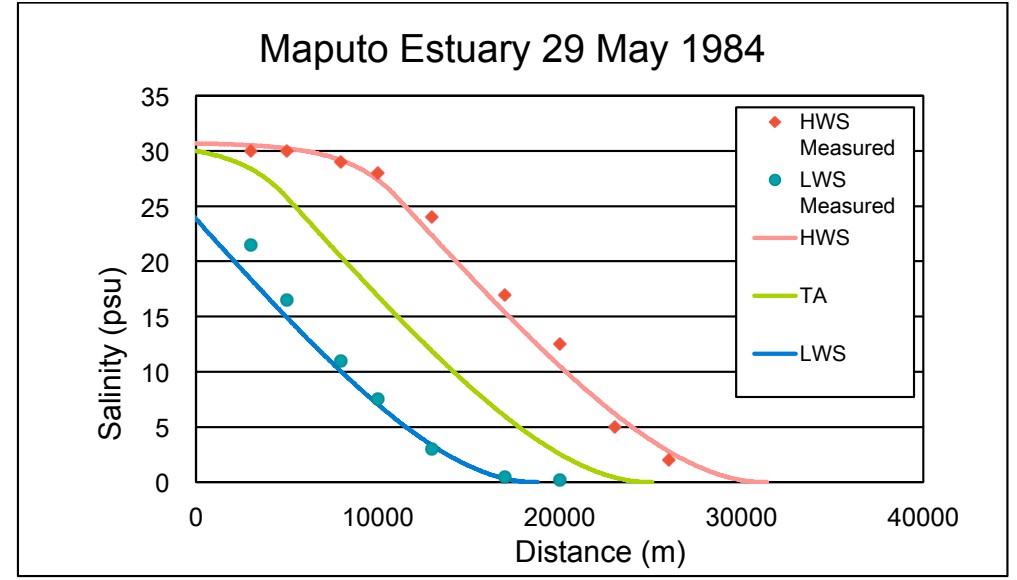

**Figure 4. Application of the numerical solution to observations in the Maputo Estuary for high water slack (HWS) and low**
**water slack (LWS). The green line shows the tidal average (TA) condition. The red diamonds reflect the observations at HWS**
**and the blue dots the observations at LWS on 29 May 1984.**

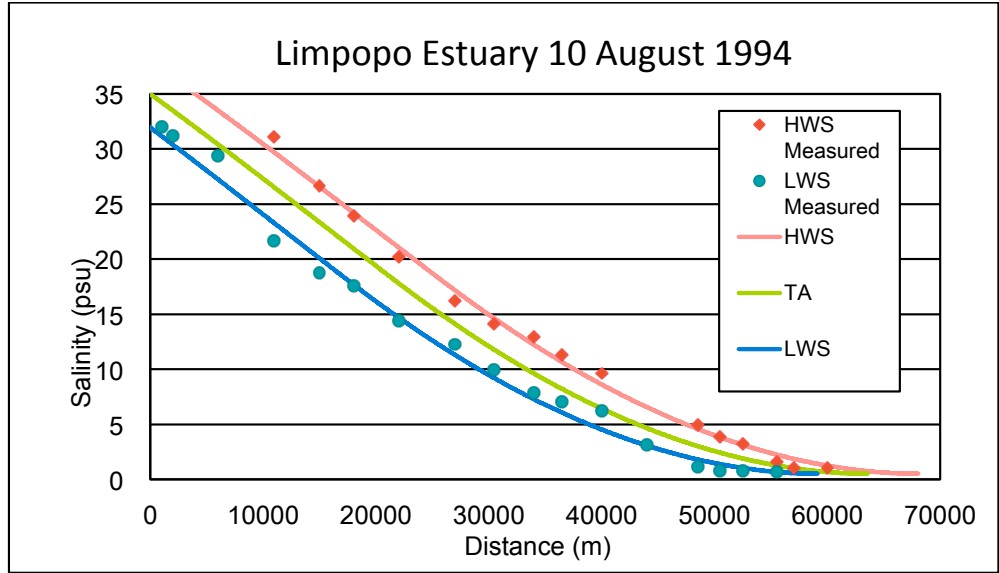

**Figure 5. Application of the numerical solution to observations in the Limpopo Estuary for high water slack (HWS) and low**
**water slack (LWS). The green line shows the tidal average (TA) condition. The red diamonds reflect the observations at HWS**
**and the blue dots the observations at LWS on 10 August 1994.**



## 4. Discussion and conclusion

Making use of the Maximum Power (MP) concept, it was possible to derive an additional equation
to describe the mixing of salt and fresh water in estuaries. Together with the salt balance equation
these two first order and linear differential equations only require two boundary conditions (the
salinity and the dispersion at some well-chosen boundary) to be solved. If the estuary has an
inflection point in the geometry, then the preferred boundary condition lies there, otherwise the
boundary condition is chosen at the ocean boundary.

This new equation can replace previous empirical equations, such as the Van der Burgh
equation, and does not require any calibration coefficients (besides the boundary conditions). The
new equation appears to fit very well to observations, which adds credibility to the correctness of
applying the MP concept to fresh and salt water mixing.

The method presented here is based on a system's perspective, which is holistic rather than
reductionist. Reductionist theoretical methods have tried to break down the total dispersion in a
myriad of smaller mixing processes, some of which are difficult to identify or to connect to
conditions that make them more or less prominent. The idea here is that in a freely adjustable
system, such as an alluvial estuary, individual mixing processes are not independent of each other,
but rather influence each other and jointly work at reducing the salinity gradient at maximum
dissipation. The resulting level of maximum power and dissipation is set by the boundary
conditions of the system. It then is less important which mechanism is dominant, as long as the
combined performance is correct. The maximum power limit is a way to derive this joint
performance of mixing processes. The fact that the relationship derived from maximum power
works so well in a wide range of estuaries, is an indication that natural systems evolve towards
maximum power, much like a machine that approaches the maximum performance of the Carnot
limit.

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
