# Peer review of "Thermodynamics of Saline and Fresh Water Mixing in Estuaries"

_Earth System Dynamics, 2017_

## Referee Comment (RC1) · Z. Wang (Referee) · 15 Nov 2017

The paper presents an extreme power theory for the mixing of saline sea water and fresh river water in estuaries. By maximizing the power of energy dissipation with respect to the dispersion coefficient an additional differential equation is derived. This equation can be solved together with the advection-diffusion equation to determine the distribution of the dispersion coefficient and the salinity in an estuary.

There are more examples of extreme power theories, e.g. the minimum stream power theory in fluvial hydraulics. They are often controversial, but they also provide good results. This is also the case with the present theory. I have some doubts about the theory, which I will explain later. However, the derived equation appears to work well. The calculated salinity distribution in various estuaries agree better with observations than the results of previous models, even though an empirical relation needed in the previous models is replaced by the theoretically derived equation. Therefore, I would like to recommend the publication of the paper despite of my doubts about the theory.

First, I have a couple of questions on the theory:

1. Is it maximization or minimization of the power? In fact the used equation $\frac{\partial P}{\partial D} = \mathbf{0}$ is only saying that the power is extreme, thus maximum or minimum.
2. Why is the kinetic energy of the river water not taken into account? In fact the total energy dissipated in the estuary is the potential energy plus the kinetic energy of the river water flowing into the estuary. Upstream of the estuary, the potential energy is dissipated by bed friction while the kinetic energy remains more or less constant.

Further I have some concern about the step from equation (9) to equation (10). First this step is a bit difficult to follow. Therefore I would like to see an more extended description of the derivation by including the used equations e.g.

$$\frac{\partial A}{\partial D} = \frac{\partial A}{\partial x}\frac{\partial x}{\partial D} = \frac{A'}{D'} \qquad\qquad (1)$$

$$\frac{\partial S}{\partial D} = \frac{S'}{D'} \qquad\qquad (2)$$

These equations are essential in the derivation but they are not obvious as they cause some concerns of mine. I am not sure about their validity. I can give a simple example showing e.g. (2) is correct: both $S$ and $D$ are linear functions of $x$. However, I can also give an example showing that such way of doing the differentiation can be wrong. It is complicated how $S$ and $A$ depends on $D$. Therefore, let take an easier example. We know that the cross-sectional area $A$ of an estuary is dependent on the location $x$ and the water level $\mathbf{h}$ which also varies with $x$. We also know that

$$\frac{\partial A}{\partial \eta} = B(x,\eta) \qquad\qquad (3)$$

In which B is the width of the estuary. However, if the similar rule is followed by doing

$$\frac{\partial A}{\partial \eta} = \frac{A'}{\eta'} \qquad\qquad (4)$$

Then we will in general not end with the correct result as given in equation (3).

Finally I have the following detailed suggestions:

1. I think it is better to draw a sloping bed in Fig.1. The figure is now showing an unrealistic extra depth at the upstream end of the estuary.
2. Please include some results of $D(x)$ as well together with $S(x)$. Please also show some results of the previous models.

---

## Referee Comment (RC2) · Z. Wang (Referee) · 29 Nov 2017

The reply of the authors clarifies many things, but I still have a couple of points:

(1) About maximum / minimum. If the higher derivatives are also zero then you can strictly speaking not conclude if there is an extreme value. Therefore I would suggest to mention in the paper that the maximization of the power is an HYPOTHESIS.

(2) About the chain rule for the derivation. Let me give another simpler example: The relation between water depth and water level in a 1D steady or tide-averaged flow. It is obvious that dD/dY=1 (D=water depth, Y=water level). It is per definition that D=Y-Z (Z=bed-level). Applying the chain rule as suggested by the authors yields dD/dY=1-(dZ/dx)/(dY/dx), and is thus not equal to 1, as it should be. My conclusion is then that

[Figure]

D and Y are both functions of x alone is not sufficient for the validity of the chain rule.

(3) Sloping or flat bed. It is also my argument that the depth is more or less constant. Therefore it is better to draw a sloping bed. Otherwise you have not not-constant depth as shown by the figure in the manuscript.

---

## Short Comment (SC1) · 29 Nov 2017

We would like to thank referee #1 for his insightful comments and the positive feedback, which we have received. Below we reply to the comments in detail.

1.  About the maximization or minimization of the power.
    It is a very relevant question whether $dP/dD$ maximizes or minimizes the power. Within the community that tries to apply thermodynamic principles to the Earth system there is also debate whether some sub-systems operate at maximum or minimum dissipation. Moreover, in this case, it is not so easy to determine whether the extreme is a maximum or a minimum. The second derivative of the power (i.e., salinity gradient, as in equation (9)) with regard to the dispersion coefficient $\frac{\partial}{\partial D}\left(\frac{\partial S\prime}{\partial D}\right) = \frac{1}{D\prime}\left(\frac{S\prime\prime}{D\prime} - \frac{S\prime D\prime\prime}{(D\prime)^2} + \frac{S\prime}{aD\prime} - \frac{(S-S_f)D\prime\prime}{a(D\prime)^2} - \frac{S\prime}{D} + \frac{(S-S_f)D\prime}{D^2}\right)$ appears to equal zero when $\frac{dP}{dD} = 0$, and this is the same for the third and fourth derivative. So it is a pity that we can't reply to the comment by a mathematic approach.

    However, it is believed that whether a sub-system tunes to maximum or minimum power depends on the degree of freedom in the system. It is assumed that when the degrees of freedom are limited, a sub-system operates at minimum dissipation, but when the system has a large degree of freedom, such as the mixing in estuaries where a myriad of different mixing mechanisms are at play, the power is maximized. Whether this is correct is not certain, but intuitively it appears reasonable. Mixing of fresh and saline water increases the entropy of a system. In an estuary there are many degrees of freedom to that effect. The salinity gradient can be depleted by gravitational circulation, tidal trapping, tidal pumping, tidal shear, and ebb and flood channel shear. These mechanisms are dominant in different parts of the estuary and at different times, but also overlap. Surprisingly, the constellation of these different mechanisms appears to function as if there was only one mechanism at work. It would not be logical that the combination of different mechanisms, all contributing to mixing, would minimize the mixing of salt and fresh water. If there is an extreme, then it should be the maximum.

2.  About the kinetic energy of the river water.
    The kinetic energy of the river water lies outside the saline domain. In alluvial estuaries, the contribution of the river flow to the tidal dynamics is minor, particularly in the saline region during low flows, when well-mixed salt intrusion occurs. In the domain of interest, the kinetic energy of the river is minor. Further upstream, where the estuary becomes riverine, there can be a residual backwater, where kinetic energy is absorbed by friction, but in the saline area, this contribution is negligible compared to the kinetic energy of the tide. The kinetic energy of the tide, in turn, is dissipated by bottom friction. The exponential shape of alluvial estuaries is a manifestation of this. In the so-called ideal estuary (with exponential width and no bottom slope), the tidal amplitude is constant along the estuary axis and there is a balance between amplification due to convergence and dissipation due to friction. As a result, the kinetic energy per unit area and the dissipation of kinetic energy per unit area are equally distributed over the estuary.

3.  About the step from equation (9) to (10).

The parameters (i.e. $A$, $D$, and $S$) in this research are all tidal-averaged, and they are a function of $x$ only. The chain rule then implies that $\frac{dA}{dD} = A'/D'$

About the example the referee mentioned, $\frac{\partial A}{\partial \eta} \neq \frac{A'}{\eta'}$ because $\frac{dA}{dx} = \frac{\partial A}{\partial \eta}\frac{d\eta}{dx} + \frac{\partial A}{\partial x}$.

About the suggestions made. In the salinity intrusion area (see the supplement of the compilation of the geometry), the depth may increase or decrease slightly, but is constant in most cases. So we think it is proper to use a flat bottom.

Below we present the comparison between the new thermodynamic model and the previous Van der Burgh model (Zhang and Savenije, 2017) including the longitudinal variation of $D(x)$ for the Maputo and Limpopo estuaries. In the Maputo the boundary condition lies at $x_1 = 5000$ (m) and in the Limpopo at $x_1 = 0$.

[Figure]

[Figure]

The values of *R square* after the regression between the simulated salinity and observations are:

| Estuary | Thermodynamic | Van der Burgh |
|---------|---------------|---------------|
| Maputo | 0.98896 | 0.98978 |
| Limpopo | 0.99331 | 0.99145 |

As we can see, the thermodynamic equation works equally well, or even a bit better, than the previous model with a constant Van der Burgh coefficient.

---

## Referee Comment (RC3) · A. Kleidon (Referee) · 4 Dec 2017

This manuscript describes the application of thermodynamics to the mixing of saline ocean water and freshwater from the river. The authors show that by maximizing the power associated with mixing, they can derive a dispersion relationship that very well compares to observed data from estuaries. I think this is a very well written manuscript, shows a very novel approach based on the energetics of mixing, and should be published after a minor revision.

Most of the derivation is easy to follow and is well explained. I have one suggestion though that I think would help to better grasp the outcome. The maximum power state is invoked here as the consequence of a flux-gradient trade-off. One difference to other

max. power applications (e.g., turbulent heat fluxes) is that the boundary conditions are fixed (in terms of Q and $\Delta$h, and hence of the potential energy flux into the estuary). So I suspect that this trade-off essentially takes place along the x-axis of the estuary in some way, but I wonder how this trade-off would look like? I am not sure how this can be done, because you basically obtain a differential equation as the solution of the maximization. It would nevertheless be nice if the authors would find a way to illustrate this trade-off and what it would mean for the salinity profiles (i.e., how would the salinity profiles look like with sub-optimal dispersion).

Minor comments:

Abstract. A concluding sentence at the end of the abstract would be nice.

line 30: "increase of the salinity over the depth" – the depth of what?

A Table summarizing the variables and their description and units would be nice. Perhaps also a Figure that shows the geometry of a generalized estuary so that it is easier for a reader from outside the field to get better acquainted with the relevant variables.

line 118: Can you please explain what the difference is between water level and water depth? Does this refer to the horizontal dashed line (for h) and the blue line for z) in Fig. 1?

line 128: S' should be explained here (rather than line 132)

line 131: perhaps point out that this is the same as eq. 2?

line150: I cannot reproduce how this equation was derived. Can you please provide more explanation, and also define D'?

line 152: The solution S'=0 would only work if S = Sf, wouldn't it? I do not think it would be a solution for S $\neq$ Sf.

line 163: It would be good to provide a reference to the Van der Burgh coefficient.

For matters of transparency, I want to disclose that I have seen and commented on a version of this manuscript before submission.

Axel Kleidon

---

## Author Comment (AC1) · 7 Dec 2017

We would like to thank referee #1 for the discussion. Below we reply to the first and third comment. The second comment about the derivation of the differential equation requires an illustration and equations, which we have put in the Supplement.

1) About the maximum and minimum. The theory in this paper follows the principle of Maximum Entropy Production which states that certain complex non-equilibrium thermodynamic systems can be successfully characterised as being in states in which the rate of thermodynamic entropy production is maximised (e.g., Kleidon, 2016). Mixing processes in estuaries continuously perform work by depleting salinity gradients. In doing so, these processes produce entropy, following the natural direction given by

the second law of thermodynamics. So we assume the power to mix is maximized to maximize the entropy production in this system.

3) About the sloping or flat bed. In fact, the slope is small so the increase of depth due to the salinity difference ($\Delta h$) is trivial compared to the depth at the estuarine mouth (h) (but $\Delta h$ is important for the mixing).

If, however, a downward slope were introduced in the picture, then we would have to include the bottom pressure as well: the water pressure over the additional depth near the downstream boundary would have to be balanced by the horizontal component of the pressure exercised on the water by the estuary bottom. This would make the sketch unnecessary complex.

Please also note the supplement to this comment:
https://www.earth-syst-dynam-discuss.net/esd-2017-87/esd-2017-87-AC1-supplement.pdf

**Supplement:**

We would like to thank referee #1 for the discussion. Below we reply to the comments in detail.

1) About the maximum and minimum. The theory in this paper follows the principle of Maximum Entropy Production which states that certain complex non-equilibrium thermodynamic systems can be successfully characterised as being in states in which the rate of thermodynamic entropy production is maximised (e.g., Kleidon, 2016). Mixing processes in estuaries continuously perform work by depleting salinity gradients. In doing so, these processes produce entropy, following the natural direction given by the second law of thermodynamics. So we assume the power to mix is maximized to maximize the entropy production in this system.

2) About the derivation.

The parameters along an estuary (i.e. $A$, $D$, and $S$) in this research are all tidal-averaged.

The relations between the parameters according to the salt balance equation, $\frac{\partial S}{\partial t} = \frac{\partial}{\partial x}\left(QS - DA\frac{\partial S}{\partial x}\right)$, can be considered as (for deriving $S'$):

[Figure]

$(S - S_f)$ was simplified by $S$.

However, when the optimum situation is achieved, it is a steady state and $S' = QS/AD$.

For calculating the parameters along the estuary in optimum situation, all the parameters are functions of $x$ only (thick line route in the above scheme):

$$\frac{dS'}{dD} = \frac{dS'}{dx}\frac{dx}{dD} = \left(\frac{\partial S'}{\partial S}\frac{dS}{dx} + \frac{\partial S'}{\partial D}\frac{dD}{dx} + \frac{\partial S'}{\partial A}\frac{dA}{dx}\right)\frac{dx}{dD} = \left(\frac{Q}{AD}\frac{dS}{dx} - \frac{QS}{AD^2}\frac{dD}{dx} - \frac{QS}{A^2D}\frac{dA}{dx}\right)\frac{dx}{dD}$$

$$= \frac{Q}{AD}\left(\frac{dS/dx}{dD/dx} - \frac{S}{D} - \frac{S}{A}\frac{dA/dx}{dD/dx}\right) = \frac{Q}{AD}\left(\frac{S'}{D'} - \frac{S}{D} - \frac{S}{A}\frac{A'}{D'}\right) = 0$$

where: $A' = \frac{dA}{dx}$, $D' = \frac{dD}{dx}$ and $S' = \frac{dS}{dx}$.

(3) About the sloping or flat bed. In fact, the slope is small so the increase of depth due to the salinity difference ($\Delta h$) is trivial compared to the depth at the estuarine mouth ($h$) (but $\Delta h$ is important for the mixing).

If, however, a downward slope were introduced in the picture, then we would have to include the bottom pressure as well: the water pressure over the additional depth near the downstream boundary would have to be balanced by the horizontal component of the pressure exercised on the water by the estuary bottom. This would make the sketch unnecessary complex.

---

## Author Comment (AC2) · 7 Dec 2017

We would like to thank referee #2 for his insightful comments and the positive feedback, which we have received. Below we reply to the major comments in detail. The minor comments will all be addressed in the final version.

1) About how the trade-off would look like.

We agree that the system is different from some other thermodynamic systems.

Let's introduce two situations shown in the following figure. The water level (blue lines) has a slope as a result of the salinity distribution (red lines), with the seaside on the left and the riverside on the right.

[Figure]

The first situation is shown by the **thin lines** and the second (optimum) by the **thick lines**. For the tidal averaged case discussed in the research, within the salinity intrusion length ($L$) to where the salinity approaches freshwater salinity, the salinity difference between the seaside ($S_{Ocean}$) to the end ($S_f$) is invariant, so the increase of depth due to the salinity difference ($\Delta h$) is constant as well (also see the hydrostatic pressure distributions in Figure 1).

However, in the first situation where $\Delta h/L$ is large (salinity gradient $S'$ is large), the salt flux from downstream $ADS'(t_1)$ at any location along the estuary is large as well, larger than the salt advection from upstream). As a consequence the salinity would increase. Hence, the salinity intrusion length increases, diminishing the salinity gradient, which would in return reduce the dispersive salt flux. Over time, the product of the salt flux and salinity gradient will attain its maximization. The tidal average salinity distribution will then not go further upstream (to the upper right quarter of the system).

The time needed to achieve the optimum situation is not sure (it could be larger or less than a tidal period). In a low flow situation (which is the critical circumstance for salt intrusion) the variation of the river discharge is slow (following an exponential decline). If the time scale of flow recession is large compared to the time scale of salinity intrusion then it is reasonable to assume that thermodynamic optimum is achieved based on the steady state assumption.

2) About the derivation in Line 150.

The comment of Referee #2 is very similar to the one made by Referee #1. We replied to it there in much detail.

For calculating the parameters along the estuary in optimum (equilibrium) situation, time derivatives are not relevant. Hence we obtain:

$$\frac{dS'}{dD} = \frac{dS'}{dx}\frac{dx}{dD} = \left(\frac{\partial S'}{\partial S}\frac{dS}{dx} + \frac{\partial S'}{\partial D}\frac{dD}{dx} + \frac{\partial S'}{\partial A}\frac{dA}{dx}\right)\frac{dx}{dD} = \left(\frac{Q}{AD}\frac{dS}{dx} - \frac{QS}{AD^2}\frac{dD}{dx} - \frac{QS}{A^2D}\frac{dA}{dx}\right)\frac{dx}{dD}$$

$$= \frac{Q}{AD}\left(\frac{dS/dx}{dD/dx} - \frac{S}{D} - \frac{S}{A}\frac{dA/dx}{dD/dx}\right) = \frac{Q}{AD}\left(\frac{S'}{D'} - \frac{S}{D} - \frac{S}{A}\frac{A'}{D'}\right) = 0$$

$(S - S_f)$ was simplified by $S$.

where: $A' = \frac{dA}{dx}, D' = \frac{dD}{dx}$ and $S' = \frac{dS}{dx}$.

---

## Author Response (AR1)

Dear Editor,

Thank you for your positive decision. After the constructive, and positive, remarks of the editors, we replied to their concerns in detail and clarified the points on which misunderstanding was raised in the discussion forum. We have made some corrections in the formulations and made some textual corrections. Also we clarified the descriptions of the figures. Finally, we added a table with notations of the parameters used. The marked-up version shows all the changed made to the original draft.

Thank you again for the review process. We hope that the final version is found in order.

Sincerely,
Hubert Savenije
Zhilin Zhang

[revised manuscript text omitted]